

# Extreme Wildfire occurrence in response to Global Change type Droughts in the Northern Mediterranean

Julien Ruffault[1,2,3], Thomas Curt[1], Nicolas K. Martin St-Paul[4], Vincent Moron[2,5] and Ricardo M. Trigo[6]

[1] IRSTEA, UR RECOVER Aix-en-Provence, France.
[2] CEREGE UM 34 CNRS, Aix-en-Provence, France
[3] IMBE, Aix Marseille Université , CNRS, IRD, Avignon Université , Marseille, France
[4] INRA, URFM, Avignon, France
[5] IRI, Columbia University, Palisades, USA
[6] Instituto Dom Luiz (IDL), Faculdade de Ciências, Universidade de Lisboa, Lisboa, Portugal

*Correspondence to*: J. Ruffault (julien.ruff@gmail.com)

**Abstract.** Increasing drought conditions under global warming are expected to alter the frequency and distribution of large, high intensity wildfires. Yet, little is known regarding how it will affect fire weather and translate into wildfire behavior. Here, we analyzed the climatology of extreme wildfires that occurred during the exceptionally dry summers of 2003 and 2016 in Mediterranean France. We identified two distinct shifts in fire climatology towards fire weather spaces that had not been explored before, and which result from specific interactions between the types of drought and the types of fire. In 2016, a long-lasting 'press drought' intensified wind-driven fires. In 2003, a 'hot drought' combining a heatwave with a press drought intensified heat-driven fires. Our findings highlight that increasing drought conditions projected by climate change scenarios might affect the dryness of fuel compartments and create several new generations of wildfire overwhelming fire suppression capacities.

## 1 Introduction

There is increasing evidence that global changes are altering the distribution and frequency of the largest, most destructive, extreme wildfires (EW) in different regions of the world (Trigo et al., 2006; Bowman et al., 2017) with ensuing economic, social and ecological impacts (Schoennagel et al., 2017). In a large number of ecosystems, anthropogenic ignitions are dominant and spatially continuous areas of surface litter and dead or live biomass are rarely limiting (Krawchuk et al., 2009; Pausas & Fernández-Muñoz, 2012). In these regions, fuel dryness therefore mainly regulates wildfires (Abatzoglou & Williams, 2016; Littell et al., 2016; Nolan et al., 2016; Fernandes et al., 2016; Turco et al., 2017). But while climate change is expected to increase the frequency and intensity of droughts in regions such as the Mediterranean (Hoerling et al., 2012), our understanding of the impact of increasing drought on EW remains incomplete, because of still unresolved uncertainties. First, meteorological drought is a complex phenomenon whose magnitude, length and timing depends on multiscale spatial and temporal interactions between evapotranspiration and precipitation (Ruffault et al., 2013). Several types of droughts can



be defined, depending on their characteristics (Hoover et al., 2015; Hoover & Rogers, 2016). "Press droughts" are subtle but chronic reductions in water availability, driven by long-term (month to seasons) reductions in precipitation and/or warmer temperatures, which increase potential evapotranspiration and reduce soil moisture (Hoover & Rogers, 2016). By contrast, "pulse-droughts" (or "flash droughts"; Mo & Lettenmaier, 2016) are short in duration (days to weeks) but extreme in

magnitude. When combined, press and pulse droughts may yield 'hot droughts' (Overpeck, 2013).

Second, translating drought characteristics into fuel moisture content is not straightforward, because fire-prone ecosystems consist of several fuel compartments that respond to drought on different timescales (Pyne, 1996; Nolan et al., 2016). Fuel moisture of trees and shrubs live foliage responds to mid-term to long-term droughts because of the control of vegetation over water fluxes (Ruffault et al., 2013). By contrast, the fuel moisture of fine, dead fuels and surface litter responds to short

term (hours to days) atmospheric dry conditions (e.g. Resco de Dios et al., 2015).

Third, large fire climatology itself (i.e. typical weather conditions associated with fires) might not be unique within a homogeneous biogeographic area. Indeed, after observing that the weather conditions associated to large fire occurrence may be different in Mediterranean ecosystems, Ruffault et al. (2016) proposed a new hypothesis wherein large fire climatology was described as the coexistent combination of a limited number multi-scalar weather conditions called Fire Weather Types

(FWTs). The FWTs are the result of the linkage between instantaneous and antecedent regional weather patterns and theirs impacts of fire behavior for vegetation communities. For instance, two main FWTs are generally observed in woody-fueled crown fires in forests and shrublands: wind-driven and heat-driven (convective) fires (Rothermel, 1991; Flannigan & Wotton, 2001; Jin et al., 2015; Duane et al., 2015; Ruffault et al., 2016, 2017; Cardil et al., 2017). Convective fires are dominated by airstreams created through convection caused by the fire which arises as a consequence of the particular

combinations of high fuel loads, anomalously warm conditions, atmospheric stratification and very low moisture content. By contrast, wind driven fires are dominated by strong winds that transport the flame closer to fresh fuel and accelerate the rate of spread.

During the summers of 2003 and 2016, Mediterranean France (see Figure 1) experienced several extreme wildfires that raised issues regarding how increasing drought conditions might affect fire activity. Indeed, despite growing efforts in fire

management and suppression capacities implemented since the beginning of the 90's (Fox et al., 2015; Ruffault and Mouillot, 2015; Curt and Fréjaville, 2017), several fires became particularly large and devastating during these two years. Besides, the 2003 and 2016 summers are also known to be particularly dry in Mediterranean France, although they did not share similar climatic characteristics. Summer 2003 combined a long drought and a summer heat-wave (Trigo et al., 2005) resulting in a 'hot drought'. Summer 2016 was marked by a long-lasting 'press drought' (French national meteorological

institute; Météo-France, 2017; Figure 2). Interestingly, these two events closely reflect the expected impact of climate change on drought that are projected in the Mediterranean (Sousa et al., 2011; Hoerling et al., 2012) and in other regions of the world (e.g. Mazdiyasni & Aghakouchak, 2015).

Following these considerations, the overarching goal of this study is to determine whether the increasing drought conditions observed during the years 2003 and 2016 are responsible for to the occurrence of extreme wildfires in Mediterranean France.





Specifically, we aimed at understanding how the weather conditions observed during these summers, although different, led to extreme wildfire occurrence. Our hypothesis is that the climatology of different types of fires have been affected by these two droughts. To answer these questions, we compared the weather and drought conditions associated to the 2003 and 2016 extreme wildfires with the historical climatology of large fires.

## 2. Material and Methods

### 2.1 Study area

The study area includes two French administrative districts (Figure 1, total area of 11,157 km$^2$) recognized as two of the most fire-prone regions in France (Lahaye et al., 2014). The climate is Mediterranean with hot and dry summers, cool and wet winters and high inter-annual variability. Although these two districts share a large number of common climatic characteristics, interannual variations in the amount of precipitation can differ greatly between them because of the existence of a west-east gradient in precipitation variability over southeastern France (Figure 2, see also Ruffault et al., 2017). Therefore, two distinct areas were considered in the following analyses (hereafter called the Western and Eastern area; Figure 1) based on the administrative borders of these districts. Fuels are similar in these two areas, consisting mostly of shrublands, pine forests, oak forests, and mixed pine oak forests (Curt et al., 2013) but the western area, which includes the Aix-Marseille Metropole urban area, is more urbanized (average population density of 394 inhabitants.km$^{-2}$) than the eastern area (174 inhabitants.km$^{-2}$) (French National Institute of Geographic and Forest Information; IGN, 2016). The average yearly burnt area during the summer season (June to September – JJAS–) from 1996 to 2016 is 1,393 ha.y$^{-1}$ and 1,389 ha.y$^{-1}$ in the western and the eastern area, respectively. Fire activity is highly variable from year to year and reached its highest level in 2003 (18,763 ha) and 2016 (4,825 ha) in the eastern and western area, respectively (Figure 3).

### 2.2 Fire data

Fire data for the summer season (June to September) were extracted from the PROMETHEE fire database (*www.promethee.fr*) for the period from 1996 to 2016. This database is managed by French forest services and provides reliable fire statistics with no major inconstancies over the period examined (Ruffault and Mouillot, 2015, 2017). For each fire, PROMETHEE provides its ignition date, size and location of fires on a 2x2 km grid. In this study, large fires were defined as fires greater than 250 ha in order to match the minimum size of extreme wildfires (EW) that occurred during the 2003 and 2016 fire seasons (see below). Wildfire events were then classified into three distinct groups: (i) extreme wildfires (EW) that occurred during the summer of 2003 in the eastern area (hereafter EW2003), (ii) EW that occurred during the summer of 2016 in the western area (hereafter EW2016) and (iii) other large summer wildfires that occurred between 1996 and 2016 (hereafter OLW).

EW were defined here as fires that caused or threatened to cause some economic, social, or ecological damages. Because of the absence of a national, comprehensive and quantitative dataset on the vulnerability of landscapes to wildfires, EW





identification was based on a systematic literature review of scientific papers, governmental reports and newspaper articles. We acknowledge that some subjectivity was unavoidable in this exercise, but as only few ignitions turn into large fires in Mediterranean France, large fires are systematically covered and well documented. In line with the detailed description of the 2003 summer fire season performed by Favre & Schaller (2004), six large wildfire events were particularly harmful and

classified as EW for the year 2003 (Table 1). For the year 2016, three wildfires were classified as EW. These wildfires were extensively reported in the media and discussed by experts across the country, in part due to their large size but also because they threatened either industrial, socio-economical or ecological values (see Table 1). In fact, the occurrence of these three wildfires is one of the main reasons why we initiated our study. Finally, the OLW group consisted of all large wildfires that were not classified in either of the two previous groups (n=29; median size = 520 ha).

**2.3 Data analysis**

The climatology of large fires was characterized by using the five following daily variables: temperature, wind speed, relative humidity and two proxies of fuel aridity. Only the weather and fuel conditions of the day of fire occurrence were analyzed because fires seldom last for more than a day in the Mediterranean. Fuel aridity was approximated through the drought code (DC) and the duff moisture code (DMC) of the widespread Canadian fire weather danger rating system

(CFFDRS) (Van Wagner, 1987). The DMC is an indicator of droughts of short duration (from days to weeks) and is generally associated with the moisture content of surface dead fuels in the Mediterranean. DMC is computed from daily rainfall, relative air humidity and air temperature during and prior to the fire day. The DC is an indicator of longer, deeper droughts (from weeks to months). It is computed from daily rainfall, relative air humidity and air temperature during and prior to the fire day. The DC is generally associated with the moisture content of slowly drying fuels such as living shoots of

plants, although the responsiveness of living fuel to weather can greatly vary among species according to their ability to dynamically adjust their water status (Viegas et al., 2001).

Daily climate records for precipitation, wind speed, temperature and relative humidity for the period 1996-2016 were obtained from climate stations. The two weather datasets were obtained from the Marignane (43.436°N; 5.215°E, altitude 5 m a.s.l.) and Le Luc (43.381°N; 6.385°E, altitude 80 m a.s.l) meteorological stations for the Western and Eastern Area,

respectively (Figure 1, source Meteo France). Both stations are located close to the center of their respective area and 75% of fires are within a 35-km perimeter of the station (Figure 4).

In addition to the five above-mentioned drought proxies and weather variables, we also examined the seasonal and inter-annual variations in live fuel moisture content (LFMC) for distinct Mediterranean shrubs species sampled from nearby wildfire locations. Historical observations of LFMC for the 1996-2016 period were obtained from the "Réseau Hydrique"

dataset, fully available in a *Zenodo repository* (Cabane et al., 2017). It provides weekly measurements of LFMC during the fire season (generally from May to September depending on the level of fire danger) for various species and sites of the French Mediterranean area. LFMC is given on dry mass basis of vegetation tissue:




$$LFMC = \frac{Wf - Wd}{Wd},$$  (1)

where Wf and Wd are the fresh and dry weights the sampled fuel material, respectively. For each area, we extracted the closest site from the barycenter of the fires under study: "Le Romaron" (43.354°N, 5.166°E, altitude 152 m a.s.l) and "Le Haras du Rastéou" (43.477°N, 6.483°E, altitude 147 m a.s.l) for the Western and Eastern areas, respectively (Figure 1,

Figure 4). For each site, LFMC was examined for two species with distinct regeneration strategies (seeder and resprouter) as they generally display some distinct functional responses to drought (Vilagrosa et al., 2014).

### 2.4 Statistical analysis

Daily weather conditions were computed for each large wildfire and averaged for each fire group. Then, to assess whether and to what extent does the climatology of the large wildfires of 2003 and 2016 differ from that of other large wildfires, fire

climatologies for between them and to mean summer daily conditions for the period from 1996 to 2016. For each weather variable, the significance of the difference in mean between these groups was tested using the Welch's modified t-test, that is more reliable than the student t-test when the two samples have unequal variances and unequal sample sizes (Welch, 1947). The daily and seasonal variations in weather conditions and fuel moisture content for the year 2003 and 2016 were also examined and compared to the mean climatic conditions.

## 3. Results

The climatology of the three different wildfires groups exhibited distinct patterns (Figure 5). Wildfires belonging to OLW were associated with significantly higher daily wind speed (Welch's two sample t-test, $p<0.001$), lower relative humidity ($p<0.001$) as well as both higher DC ($p<0.001$) and DMC ($p<0.001$) than the mean summer daily conditions for both areas (Figure 5a, 5f and 5e). This is in accordance with the dominant wind-driven fire FWT observed in Mediterranean France

(Ruffault *et al.*, 2016, 2017). By contrast, we did not observe a significant difference between the temperatures associated to OLW and the mean summer daily conditions ($p>0.1$, Figure 5a).

The climatology of EW2003 considerably differed from that of OLW. On average, EW2003 was characterized by warmer ($p<0.001$, Figure 6) and drier (DC and DMC, $p<0.01$, Figure 6) conditions than those of OLW (Figure 5a, 5e and 5f). Conversely, wind speeds associated to EW2003 were, on average, not different from the summer daily mean ($p>0.1$) but

significantly lower than that of OLW (Figure 6).

The climatology of EW2016 was very similar to that of OLW. Indeed, as OLW, EW2016 climatology was also characterized by lower relative humidity ($p<0.001$), higher wind speed ($p<0.001$), DMC ($p<0.001$) and DC ($p<0.001$) than the mean summer climatology (Figure 5d, 5e, 5f and Figure 6 ). However, EW2016 was associated with significantly higher values of DC than OLW and also than EW2003 (Figure 5c, Figure 6). Besides, it is worth noting here that the "Calanque" fire ('Cal.'

on Figure 5d) exhibited both windy and quite hot conditions (Figure 5c). This association is relatively unusual in



Mediterranean France, where strong synoptic winds, blowing from the north, are generally associated with lower temperatures than usual (Figure 5c, see also Ruffault *et al.*, 2017).

Weather conditions and live fuel moisture content exhibited strong anomalies in the month before and during the summer of 2003 and 2016 (Figure 7). These anomalies are in line with the exceptional weather conditions associated to EW2003 and

EW2016 that were described above (see Figure 5).

During the year 2016, DC was higher than usual in the western area (Figure 7a) because drought onset conditions occurred earlier this year and precipitations during spring and winter were lower than normal (Figure 7g). The DC values observed during the summer 2016 were also the highest ever recorded over the period examined (not shown). Accordingly, the fuel moisture content of living vegetation was very low during that year, for both seeders and resprouters type of shrubs (Figure

8a and 8c). By contrast, DMC and daily temperature were both close to the mean summer conditions (Figure 7c and 7e). It is worth noting that these Exceptional DC values were not observed in the eastern area (Figure 7d), which indicates the local nature of this climatic event.

During the summer of the year 2003, temperatures were particularly warm from about July 15 to August 15 (Figure 7e and 7f). Accordingly, DMC was also very high during this period and until the first half of September, especially in the eastern

area (Figure 7d). In addition, due to precipitation deficits during spring and summer (Figure 7g and 7h), DC was also above the average conditions (Figure 7c and 7d) but did not reach the levels observed in the Western area during the summer 2016. Interestingly, LFMC was generally lower than normal for both seeders and resprouters during the summer of 2003 (Figure 8a-d) but remained also higher than the levels reached in 2016 over the Western area.

## 4 Discussion

Over the next century, climate projections for the Mediterranean basin converge towards an increased severity of droughts and heatwaves, along with an elevated frequency of co-occurrence of these events (Sousa et al. 2011, 2015; Bador et al., 2017). While it is generally accepted that this warmer and drier climate will alter the frequency, intensity or severity of extreme wildfires (EW), major uncertainties remain about how it will translate into wildfire behavior and fire weather. In this paper, we analysed the weather conditions associated with the EWs that occurred during the 2003 and 2016 exceptional

summer droughts in the northern Mediterranean. On the basis of this analysis, we introduce a novel conceptual scheme summarizing the response of EW to climate-change droughts, that combines the fire weather type (FWT) framework adapted from Ruffault et al. (2016) and the drought classification from Hoover et al. (2016), which is depicted in Figure 9. According to this framework, we propose that the exceptional drought of 2003 and 2016 induced two major shifts in the meteorological "fire niche", that are expressed through some modifications of FWTs.

The first shift within the meteorological 'fire niche' is expressed through the modification of the wind-driven FWT by press droughts, as illustrated by the 2016 fire season (Figure 9). Drought was longer and more intense than usual in 2016, due to an earlier accumulation of water deficits (since previous winter, Figure 7a and 7g). Thus, although the 2016 EWs were



clearly associated with the wind-driven FWT (the dominant pattern in Mediterranean France; Ruffault et al., 2016, 2017), they were equally characterized by significantly drier-than-usual conditions. Taken together, the weather conditions associated to the 2016 EWs appear to represent a new configuration within the fire weather space (Figure 5d), in which the probability of EW occurrence might be increased. Indeed, such long and intense drought periods can lead to the substantial

level of desiccation in the living vegetation compartments, as evidenced by the significantly lower LFMC during this year (Figure 5a). This could lead to an increase in fire intensity, rate of spread and thereby reduce suppression opportunities.

The second shift within the meteorological fire niche is expressed through the modification of the heat-driven FWT by hot droughts (Figure 9), as illustrated by the 2003 fire season. Our results showed that a long-term water deficit (press drought, Figures 7b) and a heat wave (pulse drought, Figure 7f) occurred simultaneously during the summer of 2003. Several studies

documented the exceptional nature of this climatic event that occurred over central and southern Europe (e.g. Trigo et al., 2005; Ciais et al., 2005). Consistent with these observations, the 2003 EWs in Mediterranean France were associated with very hot and dry conditions but low wind speeds. While these are the standard weather conditions charactering the heat-driven FWT, such intensity in temperatures and short-term drought, that have never been observed before in the fire-weather space (Figure 5b). Such hot droughts have been identified as a trigger for to the fast desiccation of living shrubs and trees

leading to massive forest dieback (Williams et al., 2012; Allen et al., 2015) which can increase fire intensity and overwhelm fire suppression policies.

Alongside these shifts within the meteorological fire niche induced by increasing droughts, consideration should also be given to the possibility of climate change causing the emergence of new FWTs. In particular, some modifications in the interaction between large-scale atmospheric processes and regional-scale physiographic features can lead to the alteration of

the relationships between the major instantaneous weather parameters for fire spread: temperature, humidity and wind. For instance, the association of hot, windy and dry conditions is relatively unusual in Mediterranean France because strong synoptic winds, blowing from the north, are generally associated with lower temperatures than usual (Ruffault et al., 2017) (see also Figure 2c). So far, this constraint avoided the occurrence of "mixed FWT fires", combining strong winds and hot temperatures, and that are considered as one of the most critical FWT in southern latitude countries (Pereira et al., 2005,

Koutsias et al., 2012, Fernandes et al., 2016). But the weather conditions associated to the "Calanques" fire in the northern Mediterranean (Figure 5c) suggest that this type of fire should be considered as an option in the near future.

Our study suggests that 'hot droughts' and 'press droughts', both anticipated to occur more frequently throughout the XXI century according to the vast majority of models, can lead to some new generations of fires that might overwhelm suppression capacities. It underlines that the dryness of fuel is pivotal in the occurrence of EW and therefore urges modellers

to improve our estimations of fuel moisture dynamics and mortality events (see for instance Nolan *et al.*, 2016). Whether some possible changes in fire management could compensate for increasing droughts and its impact on EW occurrence remains an open and important question.





**Acknowledgments**

This work is a contribution to the Labex OT-Med (ANR-11-LABEX-0061) funded by the ''Investissements d'Avenir,'' French Government program of the French National Research Agency (ANR) through the A*Midex project (ANR-11-IDEX- 0001-02). The climatic data is obtained from climate data library hosted by Météo-France. The fire data were
provided by the Délégation à la Protection de la Forêt Méditerranéenne, France from their website at http://www.promethee.fr. The Live fuel moisture content is provided by the Réseau Hydrique, France and hosted in a zenodo repository at http://doi.org/http://doi.org/10.5281/zenodo.162978

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





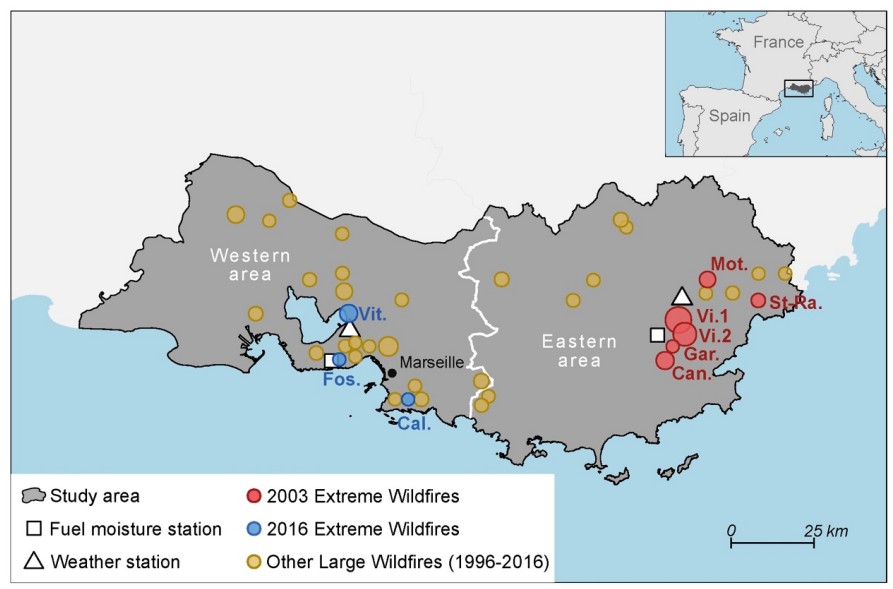

**Figure 1.** Map of the study area in southern France showing (i) the location of summer large wildfires (> 250 ha) classified into three groups, (ii) the boundary of the areas delineating different climatic patterns and (iii) the location of weather and fuel moisture stations. The size of fire points is proportional to the size of the corresponding fire (min = 250 ha; max = 5,646 ha). An individual description of the 2003 and 2016 extreme wildfires can be found in Table 1.





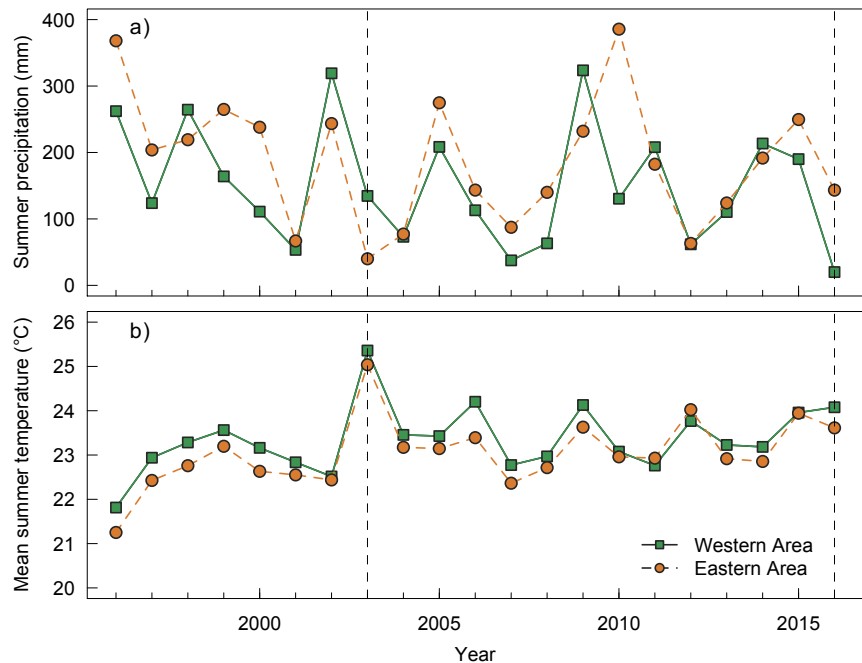

**Figure 2. Interannual variations in a) precipitation and b) mean daily temperature in the two studied areas (see Figure 1). Dotted vertical lines indicate the 2003 and 2016 years.**





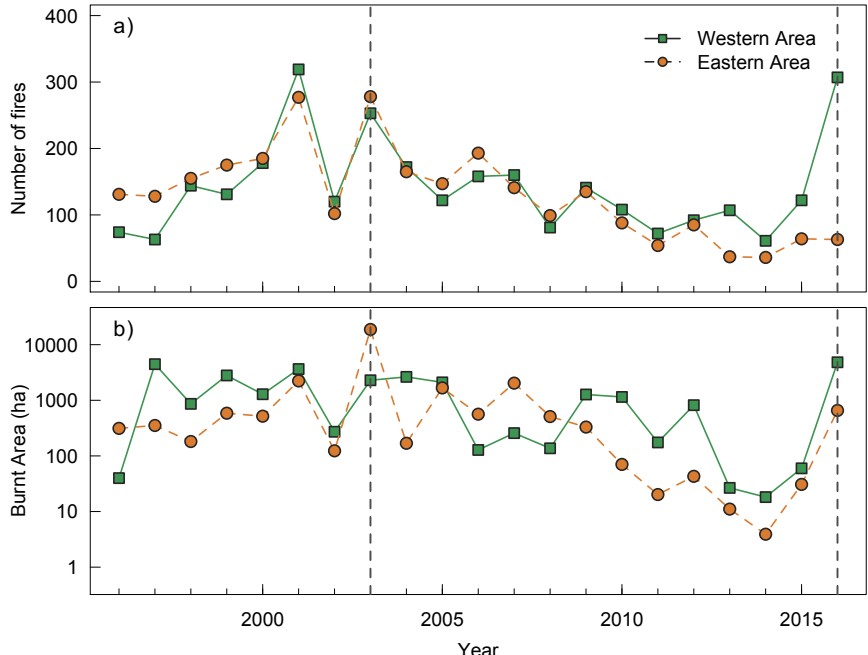

**Figure 3.** Interannual variations in a) number of fires and b) burnt areas for the two studied areas (see Figure 1). Dotted vertical lines indicate the 2003 and 2016 years.





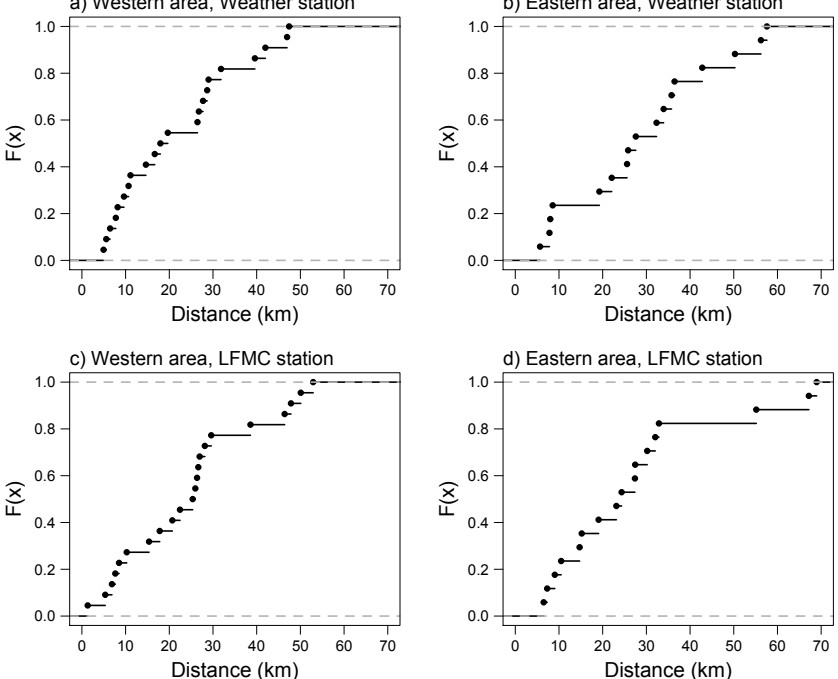

**Figure 4. Cumulative distribution function, for each studied area, of the distance between large fires their and weather stations (a et c) and of the distance between large fires and the Live fuel moisture content (LFMC) stations (b and d)**





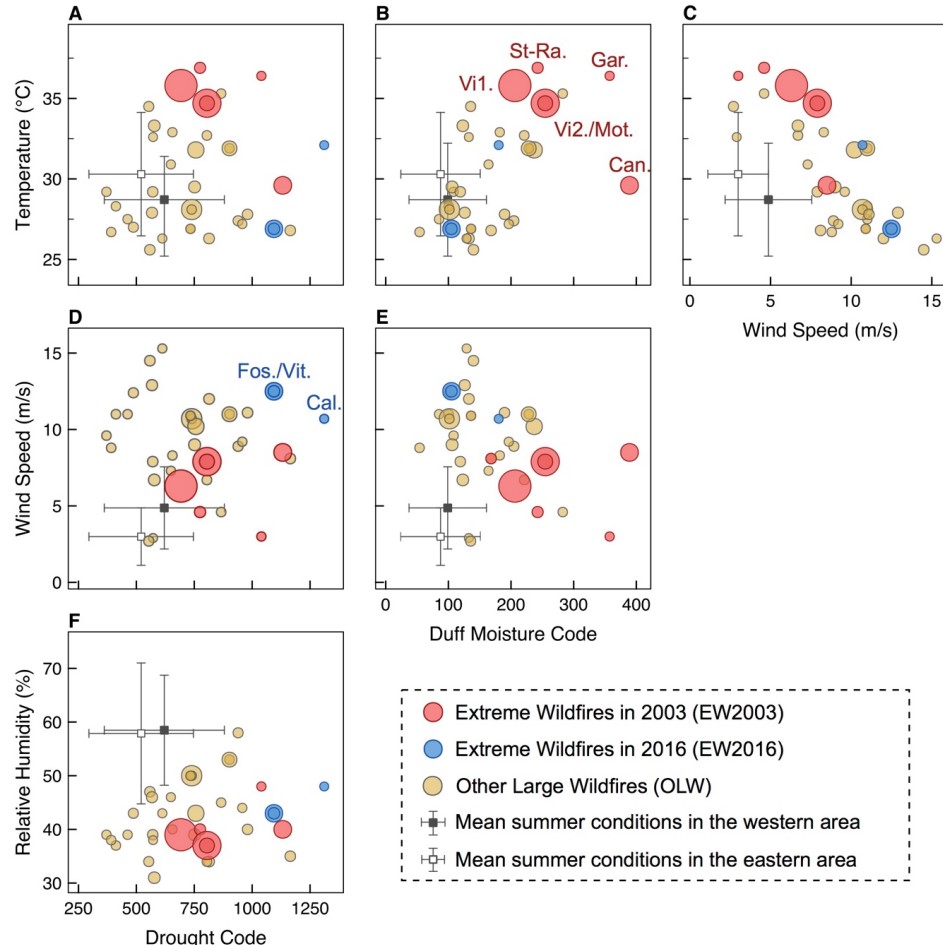

Figure 5. Bivariate plots of the daily weather and drought conditions associated with summer large fires classified into three distinct groups. The size of fire points is proportional to the size of the corresponding fire (min = 250 ha; max = 5,646 ha). The mean and standard deviations of summer weather conditions in each area are also shown. An individual description of the 2003 and 2016 extreme wildfires can be found in Table 1.





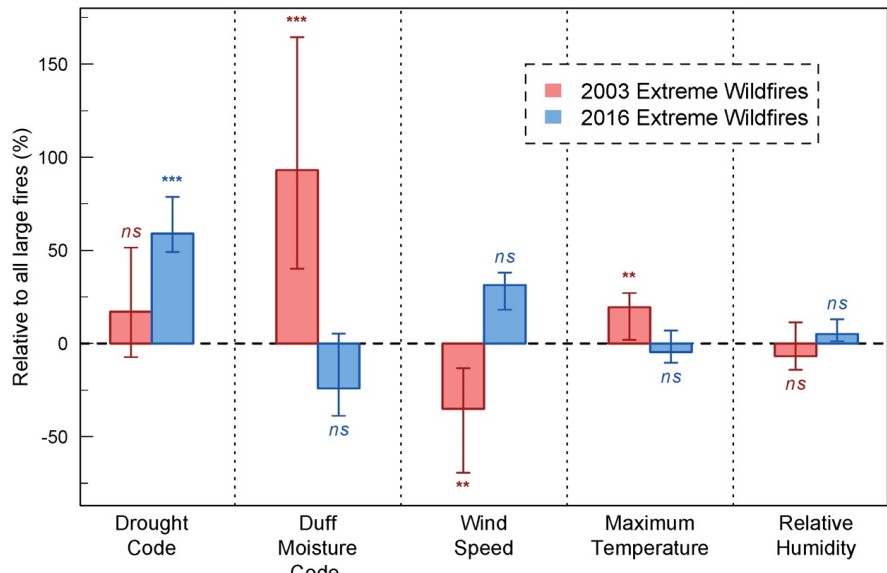

**Figure 6. Comparisons of weather conditions associated to 2003 and 2016 extreme wildfires to other large fires. Values are shown relative to other large fires for the period from 1996 to 2016. Uncertainty bars represent the minimum and maximum values. For each weather variable, the significance of the difference in mean between the extreme wildfires groups and other large fires was computed with the Welch's modified t-test (\*\*\* p<0.001, \*p<0.01, ns: non-significant, p>0.1).**





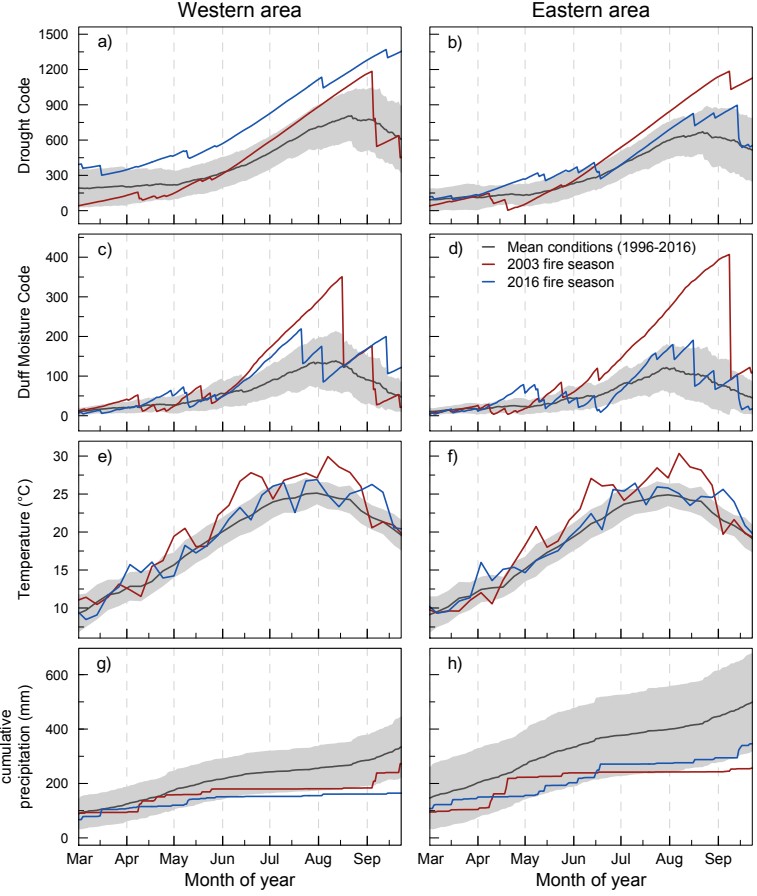

**Figure 7. Seasonal dynamics of weather conditions and drought indices during the 2003 and 2016 fire seasons in the western and eastern areas. Mean (black line) and standard deviation (shaded area) for the period from 1996 to 2016 are also indicated. All variables are shown at daily time scale expect temperatures that were averaged at weekly time scale for clarity.**



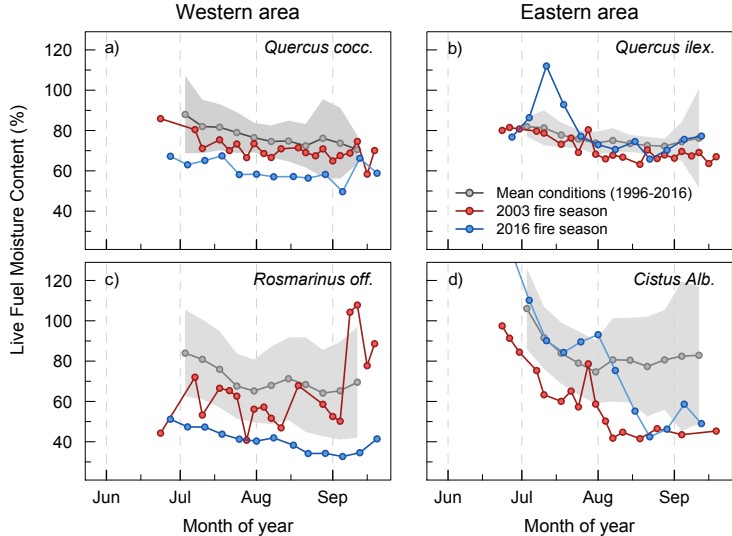

**Figure 8. Seasonal dynamics of live fuel moisture content during the 2003 and 2016 fire seasons in the western and eastern areas.**
5  **Mean (points) and standard deviation (shaded area) for the period from 1996 to 2016 are also indicated.**





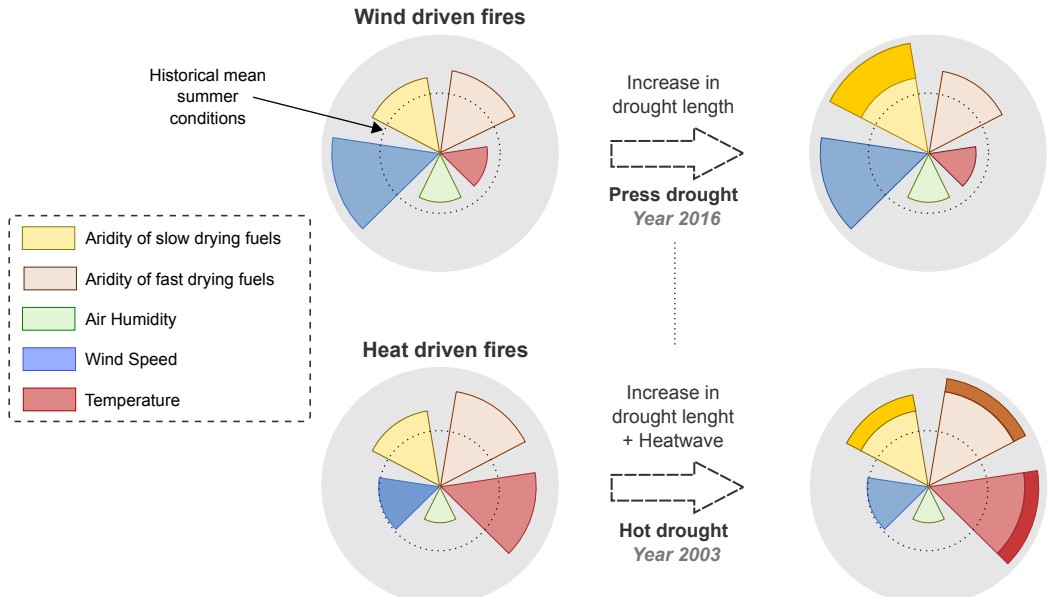

**Figure 9. Schematic representation of the emergence of two new generations of fires expected with global change type droughts in the northern Mediterranean. Two distinct shifts in the meteorological fire niche were observed and resulted from specific**
5 **combinations between the types of droughts and fire weather types (FWT). Each grey disc represents the daily weather and drought conditions associated with fire occurrence. The inner circle represents the historical mean summer daily values. Each slice represents the anomaly of the corresponding variables compared to the mean daily summer values. Panel (a) shows the modification of the wind driven FWT with press droughts as illustrated by the 2016 fire season. Panel (b) shows the modification of the heat driven FWT with hot droughts as illustrated by the 2003 fire season.**



| Name | Date | Location | Area | Size (ha) | Main type of land cover | Main damages or threats |
|------|------|----------|------|-----------|-------------------------|-------------------------|
| **Fos.** | 08/10/2016 | Fos-sur-mer | W | 2,018 | Peri-urban | Major threat to a highly sensitive industrial area |
| **Vit.** | 08/10/2016 | Vitrolles/Rognac | W | 2,663 | Industrial/Peri-urban | Several structures destroyed. Major threat to the Marseille urban area (> 1.5 M inhabitants) |
| **Cal.** | 09/05/2016 | Parc National des Calanques | W | 303 | Protected natural area | Burnt highly protected forests. Contained just before arriving in the very the touristic area of Cassis |
| **Vi.1** | 07/17/2003 | Vidauban | E | 6,744 | Peri-urban | Large camping area partly destroyed. Close to several vulnerable urbanized areas |
| **St-Ra.** | 07/25/2003 | St-Raphaël | E | 924 | Peri-urban | Close to urbanized areas |
| **Vi.2** | 07/28/2003 | Vidauban | E | 5,646 | Peri-urban | Spread close to several villages. Four fatalities |
| **Mot.** | 07/28/2003 | La Motte | E | 1,960 | Peri-urban | Three camping areas threatened (> 20,000 people) |
| **Gar.** | 08/21/2003 | La Garde-Freinet | E | 278 | Peri-urban | Close to urbanized areas |
| **Can.** | 08/31/2003 | Le Cannet-des-Maures | E | 2,728 | Peri-urban | Several structures destroyed. Three fatalities |

**Table 1. Description the extreme wildfires that occurred in Mediterranean France during the summers of 2003 and 2016.**