# Peer review of "Extreme Wildfire Events are linked to Global-Change-type Droughts in the Northern Mediterranean"

_Natural Hazards and Earth System Sciences, 2017_

## Referee Comment (RC1) · Anonymous Referee #1 · 19 Dec 2017

I read with great interest the manuscript "Extreme Wildfire occurrence in response to Global Change type Droughts in the Northern Mediterranean".

This study evaluates the climatology of extreme wildfires in Mediterranean France and identified the emergence of new generations of fires expected with global change type droughts.

Overall, the paper is interesting and it is carefully and clearly written. The conclusions are significant and can sustain the interest of a broad audience. To sum up, the authors make a convincing analysis of the problem and their manuscript deserves publication in NHESS.

---

## Referee Comment (RC2) · Anonymous Referee #2 · 1 Jan 2018

The manuscript examines extreme fire events in southern France in 2003 and 2016 and compare the fire weather conditions with those observed on average. They conclude that increased fuel dryness created somewhat novel combinations of conditions, respectively marked by pronounced fuel aridity combined with strong wind and "normal" wind combined with fuel aridity (to a lesser degree) and a heat wave. I find these conclusions sound and the ms. generally robust, although the discussion could be deeper given the wealth of information provided by the graphed results. My major concerns are some inaccuracies and errors in nomenclature (or uunderstanding of fire dynamics?), namely: - Fuel aridity does not apply to dead fuel dryness induced by short-term atmospheric influences; - What the authors describe as "heat-driven" is in

fact dryness-driven (not necessarily drought-driven, because drought implies a longer time scale).

P1, L24. Rephrase. The sentence suggests that spatially continuous fuels are expected to limite fire spread. Also simplify to "dead and live fuels", as litter is only a factor in forests.

P2, L15-16. Unclear sentence: "theirs impacts of fire behavior for vegetation communities". Do you mean "their impacts on fire behavior"?. Vegetation communities is not really needed in the sentence.

P2, L16. They burn woody vegetation, but are in fact "foliage-fueled".

P2, L16. FWT refer to fire weather types, not to fire types, so correct the sentence. Also, "heat-driven" doesn't say anything and is incorrect to refer to this type of fires. "Plume-driven" is the right designation and would be the preferred option to replace the less accurate "convective"; all fires are driven by convection as the prevailing heat transfer mechanism, irrespective of their nature or intensity. A second objection is that you have no data supporting the claim that these types of fires are convection/heat/plume driven, because that depends of atmosphere stability and stratification. All you can say is that those fires developed under weaker winds and dry conditions. Hence you should be conservative and describe them as fuel-driven or dryness-driven fires.

P2, L21. Rephrase "that transport the flame closer to fresh fuel". The flame is not transported by wind, and fuel is not "fresh".

P2, L34. Correct "for to the".

P4, L5. I understand that these fires threatened valuable resources but find hard to accept that fires of ∼300 ha are classified as extreme. The nature of the fire (i.e. extreme or not as determined by the fire environment) should not be mixed with its impact for the purpose of a study of this type. The single fact of being controlled at

such size suggests they were not that extreme. Please justify better.

P4, L13. This is not true, i.e. extreme fires can spread for several days. I suggest you justify the option by stating that the main fire runs often occur on the first day.

P4, L13-14. The concept of fuel aridity (check the original papers) is different from low fuel moisture contents attained on the short to mid-term so you should not use it in this context. Can an absence of rain for 4-6 weeks (drying time for the fuels represented by the DMC) be designated as "drought"?

P4, L16. Again, this is not correct, the fuel moisture content of surface fuels in the FWI system is expressed by the FFMC. The DMC accounts for the part of the forest floor that produces flame plus more compacted layers that smoulder, i.e. it includes sub-surface fine and coarse fuels.

P4, L19. Make the sentence clearer, because the DC was not conceived to track the moisture content of live fuels.

P5, L1-2. You certainly do not need to include this explanation.

P5, L10. This line is not understandable.

P5, L16. Wildfire, not wildfires.

P7, L12. "charactering".

P7, L26. What about the fires of 2017 in southern France? You could discuss them in this framework.

P7, L28. Not clear what you mean with "some new generations of fires". Increase in the occurrence of hot droughts and press droughts simply imply that the distribution of fire behavior characteristics will change to include more extreme fires at higher frequency.

Figure 9 is nice, but again I have problems with the nomenclature adopted (see previous comments): the aridity concept does not apply to fast drying fuels, and "heat

driven" is quite poor and inaccurate when in fact these fires are driven by fuel dryness, both on the short and long term. I suppose "heat" is used because of the high temperatures, but temperature in fire behavior only acts indirectly through its effect on live fuel desiccation and dead fuel moisture content.
* * *

---

## Author Response (AR1)

**Please find in this document our point-by-point response to the reviews and the changes that were made in the manuscript to answer these comments. Other minor changes were also made in the manuscript to improve the language and clarity. Please also note that the title also been slightly modified.**

**Reviewer # 1**

I read with great interest the manuscript "Extreme Wildfire occurrence in response to Global Change type Droughts in the Northern Mediterranean". This study evaluates the climatology of extreme wildfires in Mediterranean France and identified the emergence of new generations of fires expected with global change type droughts. Overall, the paper is interesting and it is carefully and clearly written. The conclusions are significant and can sustain the interest of a broad audience. To sum up, the authors make a convincing analysis of the problem and their manuscript deserves publication in NHESS.

We thank the reviewer for the very positive appreciation of our manuscript.

**Reviewer # 2**

The manuscript examines extreme fire events in southern France in 2003 and 2016 and compare the fire weather conditions with those observed on average. They conclude that increased fuel dryness created somewhat novel combinations of conditions, respectively marked by pronounced fuel aridity combined with strong wind and "nor- mal" wind combined with fuel aridity (to a lesser degree) and a heat wave. I find these conclusions sound and the ms. generally robust, although the discussion could be deeper given the wealth of information provided by the graphed results. My major concerns are some inaccuracies and errors in nomenclature (or understanding of fire dynamics?), namely: - Fuel aridity does not apply to dead fuel dryness induced by short-term atmospheric influences; - What the authors describe as "heat-driven" is in fact dryness-driven (not necessarily drought-driven, because drought implies a longer time scale).

We thank the reviewer for this positive appreciation and thorough review of our manuscript. Following the main recommendations, we paid particular attention to use more appropriate nomenclature when describing both fire weather types and drought features that were, when necessary, modified in the revised

text of the manuscript. We present hereafter our detailed responses to each of the comments raised by the reviewer.

(**#1**) P1, L24. Rephrase. The sentence suggests that spatially continuous fuels are expected to limit fire spread. Also simplify to "dead and live fuels", as litter is only a factor in forests.

5  Yes, this sentence has been corrected and now reads: "*In a large number of ecosystems, anthropogenic ignitions are dominant and dead or live biomass are rarely limiting…*" (P1, L24)

(**#2**) P2, L15-16. Unclear sentence: "theirs impacts of fire behavior for vegetation communities". Do you mean "their impacts on fire behavior"?. Vegetation communities is not really needed in the sentence.

'Vegetation communities' has been removed

10  (**#3**) P2, L16. They burn woody vegetation, but are in fact "foliage-fueled".

That is correct, though the expression sounds awkward. To avoid any ambiguity, the term "woody-fueled" has been removed.

(**#4**) P2, L16. FWT refer to fire weather types, not to fire types, so correct the sentence. Also, "heat-driven" doesn't say anything and is incorrect to refer to this type of fires. "Plume-driven" is the right

15  designation and would be the preferred option to replace the less accurate "convective"; all fires are driven by convection as the prevailing heat transfer mechanism, irrespective of their nature or intensity. A second objection is that you have no data supporting the claim that these types of fires are convection/heat/plume driven, because that depends of atmosphere stability and stratification. All you can say is that those fires developed under weaker winds and dry conditions. Hence you should be

20  conservative and describe them as fuel-driven or dryness-driven fires.

We agree about the reviewer's main concern: "heat-driven fires" is a misleading classification, as fire are indeed not driven by heat. As the reviewer, correctly summarized hereafter (see comment **#18**), it is very likely that the impact of temperature on fire behavior is obtained indirectly through its effect on fuel dryness. Nevertheless, we believe that the fact that very large fires occur during very intense episodes of

25  heat should be clearly mentioned. Thus, we have replaced the expression 'heat-driven' by 'heat-induced', a solution that actually enables to keep the reference to heat without misleading the readers about the main factors driving these fires. Accordingly, the paragraph describing these fire weather types has also been rewritten to frame these points clearly (P2, L18-L19).

**(#5)** P2, L21. Rephrase "that transport the flame closer to fresh fuel". The flame is not transported by wind, and fuel is not "fresh".

Yes, corrected. It now reads: "*Wind-driven fires are dominated by strong winds that accelerate the rate of spread*" (P2, L18).

**(#6)** P2, L34. Correct "for to the".

Corrected

**(#7)** P4, L5. I understand that these fires threatened valuable resources but find hard to accept that fires of ~300 ha are classified as extreme. The nature of the fire (i.e. extreme or not as determined by the fire environment) should not be mixed with its impact for the purpose of a study of this type. The single fact of being controlled at such size suggests they were not that extreme. Please justify better.

This is a relevant comment. It is our belief that the definition of Extreme Wildfire Events should not only be based on their final size but also on the direct (and indirect) economic, social and environmental impacts. Having said that, we understand that the reviewer might be surprised that a wildfire of 250 hectares could be considered as extreme while some wildfires can burn tens of thousands of hectares in some other areas, e.g. Portugal. However, in Mediterranean France where landscapes are highly fragmented and firefighting system strongly control fire size, most of the fires are small and only few fires reach extended surfaces. For instance, in Mediterranean France 43,436 fires occurred over the period 1996-2016 but only 145 were larger than 250 ha (less than 0.4%; source: *www.promethee.fr*). We agree that such information should have been mentioned on the manuscript and is now included (P4, L24-L27). The main reason why we chose to focus on 2003 and 2016 extreme wildfire events is indeed at the core of what motivated our study.

**(#8)** P4, L13. This is not true, i.e. extreme fires can spread for several days. I suggest you justify the option by stating that the main fire runs often occur on the first day.

Yes, we agree that stating that fires generally do not last more than one day in the Mediterranean is wrong. Although this statement is true for the vast majority of large fires in southern France, it can be misleading when generalized to other Mediterranean countries. We are aware that fires in Portugal (e.g. 2003, 2005, 2017) and in Greece (e.g. 2007) can last several days. Accordingly, we followed the reviewer's advice and rephrased this sentence that now reads: "*Only the weather and fuel conditions of the day of fire occurrence were analyzed because the main runs of large fires often occur on the first day in in this part of the Mediterranean*" (P4, L14-L16).

(**#9**) P4, L13-14. The concept of fuel aridity (check the original papers) is different from low fuel moisture contents attained on the short to mid-term so you should not use it in this context. Can an absence of rain for 4-6 weeks (drying time for the fuels represented by the DMC) be designated as "drought"?

That is true, we changed the term 'aridity' and adopted the more appropriate term 'dryness' throughout the manuscript, namely in section 2.3 (P4, L14) and in Figure 9.

(**#10**) P4, L16. Again, this is not correct, the fuel moisture content of surface fuels in the FWI system is expressed by the FFMC. The DMC accounts for the part of the forest floor that produces flame plus more compacted layers that smoulder, i.e. it includes sub-surface fine and coarse fuels.

Also true. This sentence has been rephrased and now reads: "*The DMC is generally associated with the moisture content of slow drying surface fuels and is computed from daily rainfall, relative air humidity and air temperature during and prior to the fire day.*" (P4, L17-L19)

(**#11**) P4, L19. Make the sentence clearer, because the DC was not conceived to track the moisture content of live fuels.

Yes, this sentence has been rephrased to clarify the primary purpose of DC. It now reads "*The DC was initially developed to estimate the soil water content of relatively deep and compacted duff and is generally associated to very slow drying fuels. It is calculated from daily rainfall and air temperature during and prior to the fire day. In the Mediterranean, DC has also been related with the moisture content of living shoots of plants, although the responsiveness of living fuel to weather can greatly vary among species according to their ability to dynamically adjust their water status (Viegas et al., 2001).*" (P4, L19-L23)

(**#12**) P5, L1-2. You certainly do not need to include this explanation.

We agreed with this suggestion. This equation was removed from the manuscript.

(**#13**) P5, L10. This line is not understandable.

Yes, we apologize for this unclear sentence. It has been rephrased and now reads: "*Then, to assess whether and to what extent does the climatology of the large wildfires of 2003 and 2016 differ from that of other large wildfires, fire climatologies were compared between them and with the mean summer daily conditions for the period from 1996 to 2016.*" (P5, L7-L9)

(**#14**) P5, L16. Wildfire, not wildfires.

Corrected

(**#15**) P7, L12. "charactering".

Corrected

(**#16**) P7, L26. What about the fires of 2017 in southern France? You could discuss them in this framework.

That is a relevant suggestion. Several very large fires occurred during July 2017 in what is named eastern area in the present manuscript (Figure 1) During the 24th and 25th of July 3 wildfires burnt more than 3,500 ha and about 10,000 residents and tourists were moved by local authorities. It seems that these fires were associated with strong winds and that the whole year 2017 was also particularly dry while spring and summer were anomalously warm (and 2017 follows an already anomalous dry and warm year in 2016). We have therefore added a sentence in the discussion to mention these fires (P7, L24-L26).

(**#17**) P7, L28. Not clear what you mean with "some new generations of fires". Increase in the occurrence of hot droughts and press droughts simply imply that the distribution of fire behavior characteristics will change to include more extreme fires at higher frequency.

The reviewer is right. Two important conclusions should have been distinguished here. In one hand and as the reviewer suggested it, an increase in the occurrence of hot droughts and press droughts will lead to extreme fires at higher frequency. On the other hand, our results suggest that droughts are also becoming more extreme, thereby leading to fire weather conditions that have not been explored before. To clarify these points, the sentence has been rephrased and now reads "*Our study suggests that 'hot droughts' and 'press droughts', both anticipated to occur more frequently throughout the XXI century according to the vast majority of models, can lead to a higher frequency of extremes wildfires and to fire weather conditions that have not been explored before*" (P7, L27-L29). A sentence in the abstract has also been rephrased (P1, L18-L19).

(**#18**) Figure 9 is nice, but again I have problems with the nomenclature adopted (see previous comments): the aridity concept does not apply to fast drying fuels, and "heat driven" is quite poor and inaccurate when in fact these fires are driven by fuel dryness, both on the short and long term. I suppose "heat" is used because of the high temperatures, but temperature in fire behavior only acts indirectly through its effect on live fuel desiccation and dead fuel moisture content.

Yes, according to the two main comments of the reviewer, Figure 9 has been modified. The expressions "Aridity" and "heat-driven" have been replaced by 'dryness' and 'heat-induced', respectively. Specific responses to can be found in our responses to comments **#4** and **#9**

[revised manuscript text omitted]

* * *

EW were defined here as fires that caused or threatened to cause some economic, social, or ecological damages. Because of the absence of a national, comprehensive and quantitative dataset on the vulnerability of landscapes to wildfires, EW identification was based on a systematic literature review of scientific papers, governmental reports, newspaper articles or feedbacks from firefighters. We acknowledge that some subjectivity was unavoidable in this exercise, but as only few ignitions turn into large fires in Mediterranean France, large fires are systematically covered and well documented. In line with the detailed description of the 2003 summer fire season performed by Favre & Schaller (2004), six large wildfire events were particularly harmful and classified as EW for the year 2003 (Table 1). For the year 2016, three wildfires were classified as EW. These wildfires were extensively reported in the media and discussed by experts across the country, in part due to their large size but also because they threatened either industrial, socio-economical or ecological values (see Table 1). In fact, the occurrence of these three wildfires is one of the main reasons why we initiated our study. Finally, the OLW group consisted of all large wildfires that were not classified in either of the two previous groups (n=29; median size = 520 ha).

**2.3 Data analysis**

The climatology of large fires was characterized by using the five following daily variables: temperature, wind speed, relative humidity and two proxies of fuel dryness on the medium (days to weeks) and long term (weeks to months). Only the weather and fuel conditions of the day of fire occurrence were analyzed because the main runs of large fires often occur on the first day in this part of the Mediterranean. The drought code (DC) and the duff moisture code (DMC) of the widespread Canadian fire weather danger rating system (CFFDRS, Van Wagner, 1987) were used as proxies of fuel dryness. The DMC is generally associated with the moisture content of slow drying surface fuels and is computed from daily rainfall, relative air humidity and air temperature during and prior to the fire day. The DC was initially developed to estimate the soil water content of relatively deep and compact duff and is generally associated to very slow drying fuels. It is calculated from daily rainfall and air temperature during and prior to the fire day. In the Mediterranean, DC has also been related with the moisture content of living shoots of plants, although the responsiveness of living fuel to weather can greatly vary among species according to their ability to dynamically adjust their water status (Viegas et al., 2001).

Daily climate records for precipitation, wind speed, temperature and relative humidity for the period 1996-2016 were obtained from climate stations. The two weather datasets were obtained from the Marignane (43.436°N; 5.215°E, altitude 5 m a.s.l.) and Le Luc (43.381°N; 6.385°E, altitude 80 m a.s.l) meteorological stations for the Western and Eastern Area, respectively (Figure 1, source Meteo France). Both stations are located close to the center of their respective area and 75% of fires are within a 35-km perimeter of the station (Figure 4).

In addition to the five above-mentioned drought proxies and weather variables, we also examined the seasonal and inter-annual variations in live fuel moisture content (LFMC) for distinct Mediterranean shrubs species sampled from nearby wildfire locations. Historical observations of LFMC for the 1996-2016 period were obtained from the "Réseau Hydrique" dataset, fully available in a *Zenodo repository* (Cabane et al., 2017). It provides weekly measurements of LFMC during the fire season (generally from May to September depending on the level of fire danger) for various species and sites of the French

[revised manuscript text omitted]